# Service Quality for Sports and Active Aging in Japanese Community Sports Clubs

**DOI:** 10.3390/ijerph17228313

**Published:** 2020-11-10

**Authors:** Tzu-Yu Lin, Seiichi Sakuno

**Affiliations:** 1Department of Gerontology and Health Care Management, Chang Gung University of Science and Technology, Taoyuan 333, Taiwan; 2Faculty of Sport Sciences, Waseda University, Nishitokyo, Tokyo 202-0021, Japan

**Keywords:** service quality for sports, active aging, age segmentation

## Abstract

(1) Background: This study aims to examine the association between service quality for sports and active aging and the impacts on different age cohorts; (2) Methods: This cross-sectional study used a convenience sample of 545 Japanese community sports club (CSC) members over 60 years old, recruited from across eight CSCs in Japan between 2012–2013. A self-administered questionnaire was used to measure the self-reported health status of the elderly, evaluations to CSCs, demographic characteristics, and information on sports programs; (3) Results: The results of multiple logistic regression showed that domains of general evaluation for sports (OR = 1.942 and 95% CI 1.336~2.824), benefits of sports (OR = 1.659 and 95% CI 1.344~2.047), and management in sports (OR = 1.273 and 95% CI 1.011~1.603) were important for club members aged 60–64, the young-old, and the old-old, respectively. With a reduced model for elderly members, stratified analyses showed a significant impact of service quality for sports on active aged people in the benefits and management domains, regardless of sociodemographic information or club participation behavior; (4) Conclusions: The findings suggest that the services encountered in sports are key to promoting health in a community. Service quality in age segmentation should be considered to promote and manage active aging in the future

## 1. Introduction

With the emergence of an increasingly aged society, healthy active aging is becoming a robust trend by which we can evaluate the quality of life in old age. Several indicators for active aging have been added to the original definition first postulated by Rowe and Kahn [1,2], such as adaptation to the changes of life, well-being, fundamental physical activity, happiness, and life satisfaction [3,4,5,6,7]. If the concept of successful aging is considered as a multidimensional but fundamental indicator of health, then active aging can be considered an advanced indicator for understanding health. The idea of active aging is to live in the present rather than dwelling upon mortality and disability; thus, embracing a positive attitude is important in aging [8,9]. Engaging in sports is a crucial factor in improving quality of life and increasing the likelihood of active and successful aging; furthermore, exercise is considered medicine/therapy, especially among the elderly [10,11]. In the past decade, numerous studies have demonstrated the mechanism and effectiveness of exercise or sports activity interventions for fall prevention, functional reablement, or cognitive performance improvement in the community or in institutions [11,12,13]. When older people participate in community sports activities, they may experience fewer difficulties going out because community clubs encourage aging in a safe and secure place, and provide an opportunity for socializing and learning among neighbors [14]. However, unstable financial support, a lack of human resources, and administrative failures result in the unstable management of community sports organizations, leading to inconsistent participation [15] or poor health outcomes [16] among the elderly. How elderly sports customers cope with their aging status through participating in sports and how they evaluate the quality of the services offered by community sports clubs remain unclear.

### 1.1. Service Quality for Sports in Communities

Several health promotion strategies have been designed to increase sports participation among older people, advocate for health-promoting activities, and support the development of national and local programs [17,18]. Although the World Health Organization (WHO) Heidelberg Guidelines for Promoting Physical Activity Among Older Persons outlined evidence for the importance of regular sports activity for older people, this study argued that barriers remain for the elderly to take the initiative regarding participation in sports [19]. The Scottish Health Survey 2017 highlighted a steep decrease in sports participation (63% to 31.5%) for the 60+ age group after excluding walking from analyses [20]. Due to limited mobility and the inertia of daily life, the elderly are inclined to age in community areas; therefore, providing quality sports services in communities becomes important. The service delivery process occurs during the transaction between the customer and service provider. Furthermore, customers make purchases in order to satisfy needs or solve problems; thus, service providers should emphasize the customer rather than the service itself [21,22]. Service is essential for generating repurchasing intention; this also applies to the sports and long-term care industries. Customers are likely to spread positive word of mouth information about a sports club following positive experiences [23,24]. Measuring the intangible quality of service by relying on customer perception is different from the tangible quality of a product, which is the predictor of customer satisfaction. The sports services that customers experience are the basis for their expectations and level of satisfaction after a sports experience [25,26]. Customers invest nonmonetary metrics such as time, physical effort, psychological effort, and sensory effort. However, if customers receive incorrect information from sports clubs during a service encounter, their evaluation of the service will be negative [27]. Community sports clubs (CSCs) should strive for customer service satisfaction rather than providing the highest quality services. For sports clubs, offering satisfying sports services for members is a crucial factor influencing satisfaction and identification. Compared with other services, sports services are unique, because physical participation is required. The intention, constraints, and benefits are likely to differ based on an individual’s sports experience, which subsequently influences their evaluation, satisfaction, and health status [28,29]. Customers who have positive experiences with services or products may be influenced by the physical environment, interactions with service providers, and the outcome. These three factors of service quality are related to high levels of satisfaction. Several studies in the service marketing literature have demonstrated that service quality and customer satisfaction result in loyalty and trust [30,31,32]. Satisfaction is influenced not only by disconfirmation expectancy, but also by factors with no relation to service quality (e.g., surroundings or personal state of mind). Satisfaction is a broad concept, whereas service quality represents specific judgments [29,33].

The best predictors of satisfaction in evaluating a healthcare service system were advanced age and self-reported health. Studies have indicated a relationship between health utilization, health outcomes, and customer satisfaction, but results show inconsistency due to the heterogeneity of age and health status [34,35]. Functional impairment influences the satisfaction of elderly health customers; in addition, staff attitudes and personal relationships are determinants of service satisfaction in a high-quality health service delivery system [36]. Advanced age is another often mentioned demographic characteristic that influences customer satisfaction among patients of hospitals and clinics [37,38]. A cross-national social survey examining the evaluation of healthcare systems in 28 European countries demonstrated a significant association between subjective health status and evaluation; healthier patients reported more positive healthcare evaluations [39]. Furthermore, Jaipaul and Rosenthal reanalyzed a regional program investigating health quality in hospitals and found similar results, indicating that older and healthier people were more satisfied with the services being offered [40]. However, the high satisfaction scores only increase between ages 65–80, after which they decline. These findings indicate the importance of age segmentation in the healthcare delivery system. The relationship among service quality, satisfaction, and health status is widely discussed in the healthcare field, but the relationship between service quality for sports and positive aging status remains unexplored in both the sports and health industry fields.

### 1.2. The Importance of Community Sports Clubs for Older Japanese People

In 2002, the Japanese government began building comprehensive community sports clubs (CSC) in all communities to create a lifelong active society and sport-friendly environment for all citizens. The main goals of the CSC project include providing convenient and affordable sports complexes for all residents in a community and creating a sports environment accessible to residents of all ages, genders, and ability levels, thus facilitating social activities. The number of CSCs has achieved great growth year-on-year a decade after the policy was implemented. The number and established percentage of CSCs in 2002, 2013, and 2019 were 541 (13.1%), 3493 (79.0%), and 3604 (80.5%), respectively (Figure 1) [41]. In terms of age, members under 12 years old constitute the largest group (19.6%); older participants include those aged 70+ years (10.9%) and adults aged 60–69 (9.5%). The number of 60+ participants will continue to rise as the population ages. Older people tend to participate in CSCs rather than other community centers and fitness clubs due to affordable prices and multiple benefits. The Japanese government advocates that CSCs should strive to provide more opportunities for people to engage in sports activities and create an intergenerational platform to bring the young and old together [42]. Previous research has highlighted the importance of intergenerational communication for older people; this relationship was considered a crucial factor for their age-related coping and social integration [43]. Furthermore, such communication improves psychological well-being and life satisfaction among older people, especially for people aged 74–85, who have significantly greater satisfaction in positive self-perception and anticipation [44]. Despite the prevalence of 60+ participants in Japan, this group, silver citizens, is not yet the primary service target, but an increasing population for CSCs [42]. Older people have access to fewer sports services than other groups in the CSCs, and it seems that they invest more than they receive. To investigate older people’s evaluation of CSC sports services, this study measured their feedback on sports service quality. Specific suggestions are provided for improving the quality of the sports environment for actively aging older people.

### 1.3. Importance of Age Segmentation in Healthy Active Aging

Trends in gerontology demonstrate the significant relationship between active aging, age groups, and gender. Demographic differences among older individuals encourage policymakers to properly attend to the needs of the aging population. Certain characteristics, including young-old age, male gender, no disability, and high education level, are most often correlated with active aging [45,46,47]; however, the relationships exist under an objective rather than subjective interpretation of successful or healthy active aging [48]. For example, women are more likely to be active agers than men in Asian countries due to the definition of social participation. Subjective age or health and self-rated health have been utilized as indicators of healthy active aging [49,50]. Gerontological studies emphasize treating older people as a heterogeneous group with different characteristics; thus, the aging cutoff is widely discussed and interpreted. Individuals in their third age, the young-old, are mostly in good physical, mental, and cognitive condition and actively engage in social activities. Individuals in their fourth age, the old-old, are inclined to suffer from chronic diseases and functional limitations, and are at high risk of social withdrawal and isolation [51,52]. Previous studies have defined the young-old as below 75 years old and the old-old as 75 years or above [53,54,55]. Old-old people experience less support from friends in comparison with the young-old due to limited mobility and a decreasing number of friends [56,57]. Previous studies focusing on well-being, enjoyment, and happiness indicated that these three dimensions generally had inverted U-shaped patterns with age, showing increased well-being up to the age of 50 years, a steep decline after the age of 50 years, and another increase after 60 years old. Conversely, negative emotions show a U-shaped pattern, with age and negative emotions (e.g., worry, stress, anger) continuously decreasing until the 80s [58,59]. The cut-off chronological age among the elderly remains inconsistent, but demographic segmentation is obviously a better strategy for improving health-related outcomes in the senior service industry [7]. Therefore, the definition of a positive aging attitude should be contingent upon the individualized aging process, and must consider gender, age cohort, living environment, and subjective health rating.

Past studies have shown a positive association between service quality and health status of the elderly in the long-term care industry, but rarely in the sports industry [25,34,35,38]. Studies have demonstrated that health outcomes are associated with different age groups. Chronological age as a predictor of health is negatively associated with physical and mental health status. In addition, segmentation for customers improves their perception of services and is essential for the provision of high-quality services [23]. However, the association between service quality in sports and a positive aging attitude stratified by age group remains unclear. This study aims to examine how age segmentation demonstrates the impacts on the relationship between service quality in sports and positive aging attitudes among senior club members in comprehensive community sports clubs in Japan.

## 2. Materials and Methods

This cross-sectional study used a convenience sample of 545 senior Japanese CSC members over 60 years of age, recruited from eight CSCs in Nerima, a ward in Tokyo, and Otsuki, a city in Yamanashi Prefecture, Japan. With the permission of the Tokyo governmental unit, this study delivered a questionnaire to five CSC leaders. Then, after completing the questionnaire, the CSCs returned it to the Tokyo Bureau of Sports, who in turn forwarded it to the author. For the three CSCs who agreed to participate, the author acted as a sports volunteer and directly observed the operations of the CSCs for a month. Then, the questionnaires were delivered at the end of each sports program the following week in CSC. The final total was 545 participants over 60 years of age from eight CSCs. The response rate was 70.32% (545/775 senior members). After two participants were eliminated due to incomplete sociodemographic or healthy aging scale assessment data, 543 participants were analyzed in this study. These two study sites were selected based on their total population, percentage of older residents, number of CSCs, and number of older CSC members. The central reason for including the Nerima district in Tokyo, with 709,262 citizens and seven CSCs [60], was that the Japanese government selected Nerima as the model for CSCs in Tokyo; thus, this district exemplifies proper practices for other Japanese sports clubs. Otsuki, with 27,505 citizens and only one CSC [61], was selected and recommended by the local government because of its population and number of older residents. The main study was conducted from July 2012–January 2013. Participant observation and self-administered questionnaires were conducted as the main source of data. This study was approved by the Academic Research Ethical Review Committee at Waseda University in Japan (#2012−022).

### 2.1. Outcome Measurement

#### Positive Active Aging Scale

A positive and active aging scale, developed to measure the self-reported health status of older people, evolved from the Japanese version of the Philadelphia Geriatric Center Morale Scale [62]. An 11-item version of the scale was developed to adapt the questionnaire for relevance to CSC culture, thereby assisting the Japanese senior club members in completing it. This scale has good internal consistency and validity, with a Cronbach’s α coefficient of 0.81 and confirmatory factor analysis of the revised scale (RMR = 0.04; GFI = 0.93; AGFI = 0.88; DELTA2 = 0.91; CFI = 0.91; RMSEA = 0.09) [63] (Appendix A). The scale contains four dimensions: psychological health, social health, positive attitude, and morale. Participants rated their agreement with experiencing each situation during the past four weeks. Each item was scored using a five-point Likert-type scale, ranging from strongly agree (5 points) to strongly disagree (1 point). The instrument score was calculated based on the sum of the 11 items. A higher score than average (48.88) indicated a much healthier aging status. The dependent variable of this study was dichotomized into health and active aging (HA, means of score ≥ 43.94) and no HA (means of score < 43.94).

### 2.2. Study Variables

#### Service Quality for Sports Scale

The independent variable, service quality for sports scale (SQS), was measured to investigate the evaluations of senior club members. The SQS scale has good internal consistency and validity, with a Cronbach’s α coefficient of 0.92 and confirmatory factor analysis of the revised scale (RMR = 0.03; GFI = 0.94; AGFI = 0.90; DELTA2 = 0.95; CFI = 0.95; RMSEA = 0.08) [63]. An 11-item version of SQS included three items about benefits, two about access, three about interactions, three about management, and two about general evaluation (satisfaction with the service and ability to meet their sports needs). Participants evaluated the sports experience at a CSC. Each item was scored using a five-point Likert-type scale, ranging from strongly agree (5 points) to strongly disagree (1 point). The instrument score was calculated by adding the scores for each domain.

### 2.3. Confounding Factors

Confounding factors included sociodemographic characteristics and information on sports programs. The sociodemographic characteristics considered were gender, age group, educational level, living arrangements, living area, and employment status. The age group was based on the participant’s age at the time of enrollment in this study. Living arrangements were defined based on other members of each participant’s household; possible classifications included living alone, with a married partner, with two generations, etc. Living area and employment status variables were both dichotomized into rural/urban and employed/not employed, respectively. In addition, information on sports program variables included length of memberships, participation frequency, number of club memberships, travel time to CSCs, and time of visiting the CSCs. Sample questions included “How often do you participate at the CSC?” and “How long is the trip from home to the CSC?” The length of membership was calculated from the date when participants enrolled in the CSC. The number of club memberships was defined by asking participants “In how many CSCs do you participate?” and dividing the answer into 1 (current CSC only) and 2+ (more than one CSC).

### 2.4. Statistical Analysis

Categorical variables were expressed as a frequency and percentage, while numerical variables were displayed using mean and standard deviation. A chi-square test was carried out to examine the categorical variable and HA group differences. An independent sample t-test was conducted to examine the bivariate association between the HA group and numerical or categorical variables. A multiple logistic regression analysis was performed to determine the association between HA and the study variables, including sociodemographic information, information on sports programs, and SQS. The model-selection strategy of the multiple logistic regression analysis was based on univariate tests and backward selection. Multivariate-adjusted odds ratios (ORs) with 95% confidence intervals (CIs) were analyzed. Furthermore, based on different age groups, stratified analyses were presented to moderate the association between HA and various factors, including sociodemographic, information on sports program, and SQS variables. A two-sided *p*-value of <0.05 was considered statistically significant. These analyses were performed using SAS for Windows, version 9.4 (SAS Institute Inc., Cary, NC, USA).

## 3. Results

A total of 543 participants were included in the analysis, of whom 293 (53.96%) were classified as HA. The average score on the healthy aging scale was 43.94. Variables of gender and age group were not significantly different between HA group. The HA group primarily included women (*n* = 218, 55.19%), people 60–65 years old (*n* = 101, 56.74%), college graduates (*n* = 95, 60.13%), those living alone (*n* = 52, 65.00%), and those living in rural areas (*n* = 62, 59.62%) (Table 1). Peak attendance time for all participants was between 9:00 a.m. and 12:00 p.m. (*n* = 372, 68.51%); in addition, favorite attendance time for HA group was between 12:00 p.m. and 3:00 p.m. (*p* = 0.039). The healthy agers were also associated with other factors related to CSC sport participation, including longer club memberships (3.57 ± 3.02), more frequent participation weekly (*n* = 33, 67.35%, *p* = 0.037), more than one club membership (*n* = 138, 62.73%, *p* < 0.001), and closer than 10 min to the CSC (*n* = 125, 56.31%), compared to the opposite group. In addition, more than half of the HA group was positively associated with sports service satisfaction (*n* = 255, 57.95%, *p* < 0.001). The HA group had a significantly higher score of SQS in benefits, access, interaction, and management domains, compared to the no HA group (*p* < 0.001) (Table 1).

For all senior members, a multiple logistic regression analysis demonstrated that living arrangements, educational level, number of club memberships, SQS-benefits, and SQS-management were significantly associated with HA (Table 2). Living alone was more likely to be HA than living with 2+ generations (OR 2.385; 95% CI 1.282–4.439). Regarding education level, seniors who had graduated from college had the greatest likelihood of HA (OR 2.331; 95% CI 1.147–4.739). Regarding information on sports programs, having more than one club membership was 1.68 times more likely to be HA (OR 1.680; 95% CI 1.123–2.514). SQS-benefits and management domains were significantly associated with HA (Model I). A reduced model with elderly members over 65 years (*n* = 365) and the significant variables in Model I demonstrated similar results (Model II). A slightly higher association was observed between HA and living arrangements/number of club memberships in Model II than in Model I based on stratified age groups. Age showed impacts on HA. Table 3 analyzed the data stratified by age groups: 60–64 (Model I), 65–74 (Model II), and 75+ (Model III). Living arrangements, participation frequency, number of club memberships, and SQS domains were significant factors associated with HA. SQS domains had significantly positive impacts on HA according to the age cohort category (Table 3).

Stratified analyses showed the significantly highest HA scores among elderly participants over 65 years with the highest SQS scores, regardless of sociodemographic information or club participation behavior (*n* = 365). For the SQS-benefits domain, the highest HA scores were found in participants living alone who fell into the higher SQS-benefits score (score ≥ 12.55). A less obvious relationship between HA scores and SQS-benefits was observed among participants living with 2+ generations. The highest HA scores were observed among participants with higher SQS-benefits scores, regardless of their education status and number of CSC memberships (Figure 2). The model of SQS-management domains (score ≥ 12.76) and the significant factors related to the HA group also showed similar results.

## 4. Discussion

CSC memberships reached 3604 CSCs during 2012–2019. The development of policy implementation showed a growth trend in 2002–2012 but a slowdown in 2014–2019. The impact of policy on sports participation requires medium-term analysis to observe the change in individuals and communities in a five-to-ten-year period [64]. Therefore, the data collection in 2012–2013 not only represents the situation of CSCs in both study sites in Japan, but, more importantly, provides feedback for the decade-later impact of sport participation in communities, indicating a significant association between high-quality sports services and active aging for senior CSC members. Lin and Sakuno (2015) demonstrated how community sports environments provide sports services for senior customers and how older people engage in sports activities with an emphasis on urban–rural differences. The results suggest that the best service provider in communities, CSCs, can deliver quality sports services for a growing segment of the population [65]. To the best of our knowledge, and based upon an extensive literature review, this study found an innovative approach to examining the association between sports management and gerontology. Furthermore, Shimizu and Yanagisawa (2015) investigated that 47.9% of CSCs in total developed in either a maintaining or shrinking trend in 2014, as evaluated using the predictor of memberships, financial resources, and number of employees and sports instructors [66]. The results showed that the stable organization evolutionary growth of CSCs achieved a mature stage, which refers to a plateau phase of the growth curve.

In this community cohort study, the proportion of subjective HA was 53.96%, which resembled that reported in a previous study conducted in European countries (56.2%) and worldwide (0.4%–95%) [45,67]. In the present study, factors including living alone, being a college graduate, having more than one club membership, and highly evaluating service quality for sports were associated with HA. This community-based study confirmed the positive association between SQS and HA; in particular, general evaluation of sports (OR = 1.942), benefits of sports (OR = 1.659), and management of sports facilities (OR = 1.273) are respectively important for senior club members aged 60–64, the young-old, and the old-old. These results also elucidated the effects of interaction by analyzing the combination of SQS domains and living arrangements, education level, and number of club memberships. The findings of the present study partially resemble and confirm some mechanisms published in previous studies that have demonstrated the relationship between service quality in the long-term care industry and social participation or physical effects among older people. The present study is one of the first to confirm the associations between HA, SQS, and the interactive effects of SQS and demographic factors on senior club members at a community sports club through primary data analyses.

### 4.1. Characteristics of Healthy Active Agers at CSCs

The demographic background results displayed the age segmentation of members. Concerning the core of active aging, previous studies have highlighted the importance of habitual sports and exercise participation as an indicator of health and active aging. In this study, over half of senior members (53.96%) exhibited healthy active aging through sports participation at CSCs. Regular sports activity can reduce the risk of physiological deficits and chronic disease, improve general well-being and mental and cognitive health, and enhance productivity and social integration [17,68]. Among all sports programs, aquatic aerobics programs, gymnastics, walking, and table tennis are the most frequently offered by CSCs according to seniors’ needs. Senior members choose their favorite sports programs based on personal interests and physical condition to improve flexibility and cardiovascular function; hence, the gymnastics program, which uses balance balls and bands to stretch, and the aquatic aerobics program, which involves walking and performing stretch gymnastics in the water, are CSC favorites, especially in Nerima, Tokyo [65]. Previous studies have highlighted the benefits of water-based exercise in decreasing hypertension and cardio-related chronic diseases [69,70]. Nevertheless, due to cultural differences, these findings are inconsistent with Western studies of sports participation for the elderly; in Western countries, the top five sports for the elderly consist of golf, tennis, bowling, track and field, and swimming [71]. Walking programs and table tennis are popular for senior CSC members in Japan. Walking is one of the most popular physical activities in Japan, for reasons related to the social environment, accessibility, and health [72]. A study investigating sports participation in Japan indicated that the top three physical activities among people in their 60 s were walking (63.8%), muscle training (13.0%), and gymnastics (15.9%) [73]. Walking programs benefit not only the functional fitness of senior members, but also support their understanding of the cultural, historical, and natural features of their native environments [72,74]. CSCs take member needs into consideration when designing their programming. The relative safety of walking on a trail or street was presented as an explanation for their popularity. Participants were enthusiastic about practicing their table tennis skills and belonging to several different clubs. Older people participate in sports to recover physical function, improve stamina, and enhance sports skills.

### 4.2. Age Segmentation in Service Quality for Sports and HA

The results of this study showed a strong relationship between service quality and healthy aging (Table 2). A higher evaluation of SQS was related to more healthy aging status. Gronroos defined service quality as “what the customer is left with when the production process is finished.” [21] With sports, outcome quality is related to each participant’s perceived experience of sports and exercise participation; they expect to gain physical, psychological, or social health benefits from sports. If sports programs do not benefit members as expected, they might evaluate the service negatively [29]. Therefore, service quality is strongly related to healthy active aging, which refers to the outcome that members obtain from sports and CSCs.

Although increasing numbers of elderly customers are engaging in sports and health activities in their communities, few studies have focused on service quality in sports, particularly for people over 60 years of age [75]. For senior members aged 60–64, the general SQS domain, evaluating good service that meets their needs/demands, is significantly associated with a positive aging attitude. Staying healthy through sports activities is the vital sports need for this cohort, similar to other members, along with participating in high-quality sports programs with good service, leading to customer satisfaction, repurchase behavior, and commitment to the sports organization [75]. Elderly members tend to positively evaluate their favorite sports programs, which increases the likelihood of their continued participation. This explains the fact that although customers may not experience the highest quality service for a variety of reasons (e.g., access, price, or availability), they may still favorably evaluate their service experience. Female participants (*n* = 142) account for 79.78% of senior members (*n* = 178). Many female members formed friendships through the CSC and socialize after their sports programs. Participating at CSCs made senior women feel they were contributing to their community. Liechty indicated that even though leisure constraints are often related to increases in age, this is not true for females [76]. Older women innovate leisure activities following their reduction in family responsibilities and build new social roles that allow them to attend to their own needs. Thus, recent research, including the present study, indicates that older women are critical in the development of community CSCs.

The results displayed the age segmentation of members and differences in evaluating CSCs. The majority of CSC senior members are participants aged 65–74, known as young-old participants (*n* = 269, 49.54%). They are primarily motivated to attend CSCs to gain benefits from sports, resulting in disease prevention and decreases in health expenditures. The benefits SQS domain included health promotion, relaxation, and friendship establishment, which are, unsurprisingly, associated with a positive aging attitude for the young-old participants. These individuals were in the discovery and innovation stage and still had the energy to engage in leisure and sports activities. The average retirement age of Japanese laborers is 65 for men, and 60 for women [77], so many start their “third age” without any life plans. They usually participate in various sports and attend leisure organizations or clubs in the neighborhood due to the availability of leisure activities that they did not previously participate in. The young-old who rated themselves as healthy had more confidence in their ability to complete the movements or actions directed by instructors; thus, they gained self-efficacy through sports experience. When older people are positive, optimistic, independent contributors to society, they adapt better to aging. They feel better when they are satisfied with life, focus less on unpleasant things, and come to accept biological aging.

An Australian national cross-sectional survey investigating sports and recreational participation showed that individuals aged 80+ (OR = 1.48; 95% CI 1.29–1.70) were more likely to participate in organized sports activities in friendly sports environments, compared to participants in the 65–69 group [78]. The management SQS domain included the quality of facilities and equipment, safety in sports, and satisfaction with the overall CSC management. Old-old participants are aware of safety concerns while participating in sports due to their physical limitations and poor health status; therefore, most studies have indicated that sports participation in indoor, safe, sedentary or low-intensity physical activities is a commonly observed phenomenon for them [79]. CSC managers consider risk-response skills, professional skills, and members’ needs when hiring coaches and instructors and designing sports programs. When conducting sports programs, the entire staff is concerned with paying attention to older members’ health conditions during exercise. A well-managed CSC gives rise to trust and satisfaction from old-old participants, which results in good outcomes and healthy active aging.

### 4.3. Effect of Demographic Segmentation and Service Quality for Sports on Healthy Active Aging

One primary goal in this study was to explore the strength of association between service quality for sports and confounding healthy active aging attitude factors. Our findings have revealed a high strength of association among demographic factors such as living alone, having graduated from college, and having more than one club membership and both management and benefit SQS domains on healthy active aging. Interestingly, our findings showed that individuals living alone who more highly evaluate the benefit of sports have a 7.74 times greater likelihood of healthy active aging than elderly participants living with 2+ generations with a lower evaluation (Figure 2). Elderly participants living alone participated at CSCs to combat the negative impacts of living alone, including isolation, loneliness, and depression [71]. They seek social connection through CSCs, with a “use it or lose it” attitude toward aging [80]. Older people were inclined to participate in sports organizations where they can build connections. Previous research has demonstrated that participating at CSCs fosters social connections and increases social network size, particularly for older people [81]. Importantly, social network size and number of social contacts were revealed as determinants of life satisfaction and successful aging among older people [43,82]. A systematic review study has highlighted that community-based sports clubs benefit older participants in negotiating a negative aging process, fostering social connections, and decreasing social loneliness, especially in rural areas [19]. The findings demonstrated that even the elderly living alone can still fight stereotypes of aging, like weakness, through sports participation. The desire to compete and perform is the motivation for active agers to participate at CSCs; therefore, the value of a well-managed CSC with good facilities and a safe sports environment to encourage active aging and self-efficacy for senior customers must not be underestimated [71].

Demographic factors such as age, gender, marital status, and socioeconomic status (SES) are the most common determinants associated with or influencing people’s general health. SES, which encompasses education, wealth, and occupation level, is of high relevance for self-reported health status and life expectancy [83,84,85]. Numerous studies have shown that SES affects one’s health outcomes, and aging will continue to accumulate these disadvantages over time [86,87]. The relationship between aging velocity and SES on health outcomes has been shown to influence different age groups on a long-term basis. These findings are consistent with previous studies demonstrating the long-term impacts of SES as a determinant of subjective health outcomes on an individual’s life transition and choices [88,89]. Elderly people with high SES were inclined to participate in organized sports clubs with abundant sports resources and facilities, rather than public sports grounds. When elderly members perceived high-quality service in the management and benefits domains, this, in turn, directly resulted in customer satisfaction, continuous sports participation, and positive health outcomes [77]. On the other hand, price is the primary determinant for the elderly to participate in community sports clubs. A Japanese government investigation of fitness and sports attitudes indicated that people hoped that private sports clubs, such as fitness clubs or swimming clubs, would decrease their fees (49.3%), improve accessibility (22.4%), and provide specific sports facilities for elderly or disabled people (13.8%) (Cabinet office, 2013). A CSC is a nonprofit organization primarily supported by the government; however, high SES elderly members are more active in sports participation and adherence due to their sedentary lifestyle before retirement and ability to participate in multiple activities, as opposed to disadvantages groups [90].

One further noteworthy result involves the number of CSC club memberships. Senior members generally go to CSCs to exercise on weekday mornings, but they still have opportunities to participate in other clubs, if available. Sirven and Debrand revealed that social engagement enhanced participant health status and highlighted the positive relationship between membership numbers at social or sports clubs and self-reported health outcomes [67]. The Japanese government established an administrative unit, the wide-area sports center, to guide and assist CSCs in support of proper policy implementation. In some special conditions, CSCs offer sports programs under government contract and receive government subsidies. In other words, CSCs are commissioned to enact government sports policies with potential financial support. If CSCs have any operating problems (e.g., with stadiums, tools, human resources, etc.), the government and the wide-area sports center are responsible for eliminating such issues. Service quality for sports is well-controlled by the government. CSCs receive financial support not only from the local government, but also from other foundations. They also cooperate with other sports organizations, such as CSCs, sports alliances, and professionals. These results explained the outcome of sports policy leading to the sports alliance in communities, which brings networks of sports resources and opportunities to the elderly for the benefit of their health. Alliances between CSCs in nearby communities would increase available resources for senior members and the clubs themselves. Participants would benefit from more opportunities to exchange sports information and connect with others with similar interests. The increasing size of social networks from sports CSCs also fosters social connections and engagement for active agers.

This study has several limitations. First, it is only a cross-sectional study with a purposive convenience sample of 545 participants in Tokyo and its neighboring area, which may limit the interpretability of the causality between service quality for sports and healthy active aging. Our findings confirmed the stratified effect of age segmentation on the association between service quality for sports and healthy active aging, but the representation is still warranted. Second, the study focused on Tokyo and Yamanashi prefecture in eastern Japan, which raises the issue of the findings’ generalizability. Thus, future studies should examine the active aging model and service quality in other Japanese communities and in other Asian countries with similar sociocultural backgrounds. Finally, this study utilized a self-designed questionnaire to represent healthy active aging, rather than using an objective assessment tool such as senior fitness tests or history of chronic disease, which may have improved the accuracy. Furthermore, it is necessary to establish a model of active aging among Asians using both qualitative and quantitative approaches and develop a supportive sports environment for the elderly.

## 5. Conclusions

This study highlighted the importance of service quality for sports provided to senior members to promote healthy active aging. The association between SQS and demographic variables on healthy active aging is confirmed in this study. Most importantly, providing high-quality sports services within age segmentation according to the needs of physical aging or sports expectations is critical to the senior, young-old, and old-old elderly in the health and sports industry. The growth of well-managed community sports organizations plays an essential role in improving elderly residents’ access to health and sports participation. Consequently, this study suggests that CSC managers or future studies consider the uniqueness of the senior customer when delivering sports services. The application of the service encounter in sports is a key trend for future active aging study. Designing programs with opportunities for interaction and communication during exercise would be optimal to decrease disadvantages associated with aging. To summarize, service quality is strongly related to satisfaction and healthy active aging; therefore, segmenting the elderly into heterogeneous customer types is necessary for future planning.

## Figures and Tables

**Figure 1 ijerph-17-08313-f001:**
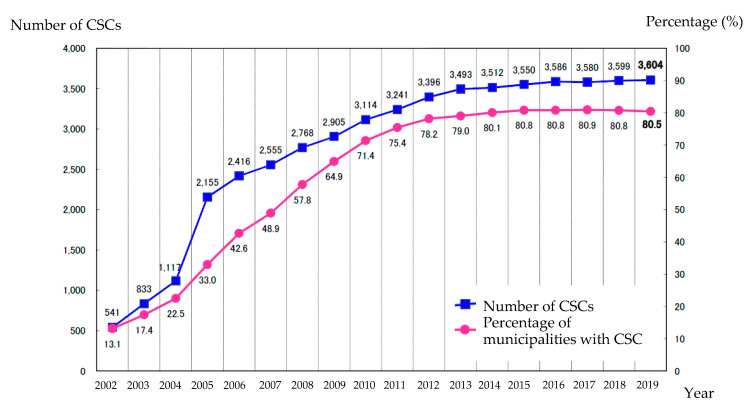
Development of community sports club in Japan, 2002−2019.

**Figure 2 ijerph-17-08313-f002:**
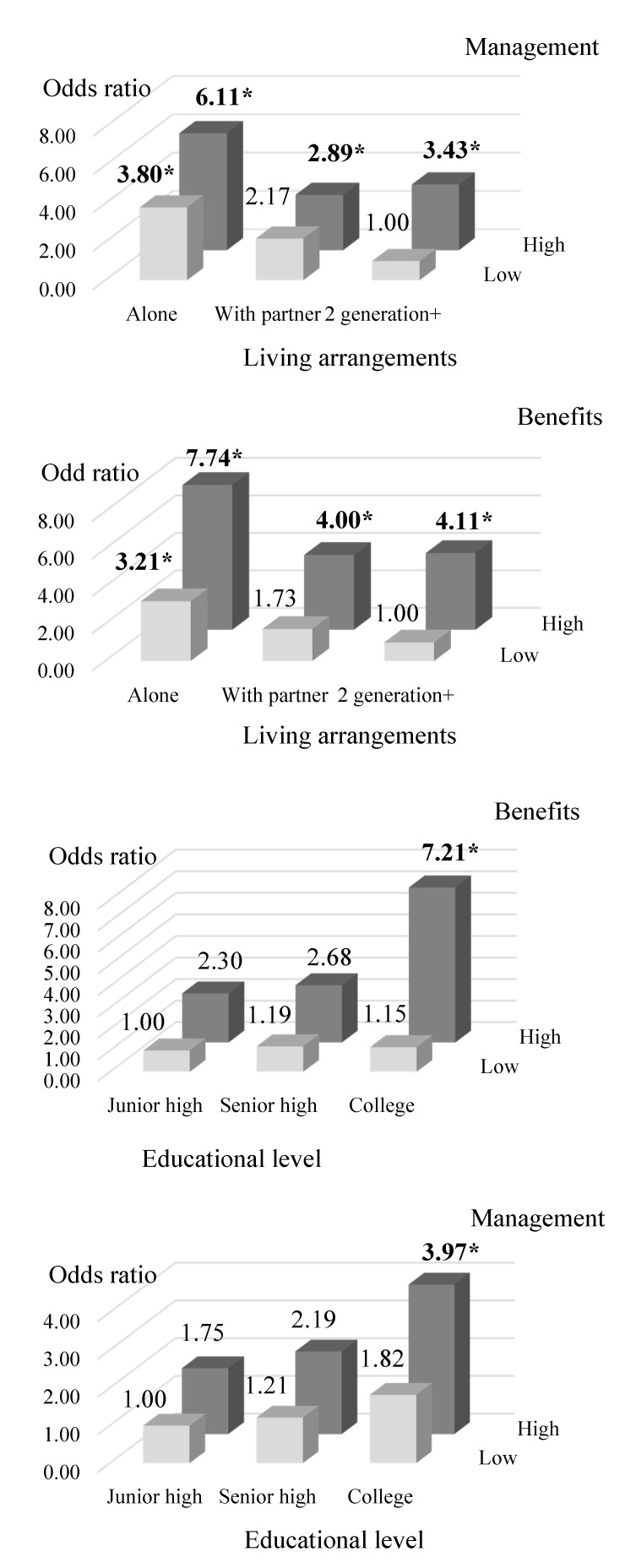
The association between sports service evaluation and healthy agers over 65. * *p*-value < 0.05.

**Table 1 ijerph-17-08313-t001:** Sociodemographic characteristics of club participants stratified by healthy active aging group.

	Total *n* = 543		Healthy Active Aging				*p*-Value
			No *n* = 250, 46.04%		Yes *n* = 293, 53.96%		
	Number	%	Number	%	Number	%	
Positive and active aging score ^※^	43.94 ± 4.76		39.82 ± 3.54		47.23 ± 2.53		**<0.001**
Gender							0.399
Women	395	72.74	177	44.81	218	55.19	
Men	148	27.26	73	49.32	75	50.68	
Age group							0.657
60–65	178	32.78	77	43.26	101	56.74	
65–75	269	49.54	128	47.58	141	52.42	
75+	96	17.68	45	46.88	51	53.13	
Educational level							0.252
Junior high	60	11.05	31	51.67	29	48.33	
Senior high	310	57.09	150	48.39	160	51.61	
College	158	29.10	63	39.87	95	60.13	
Other	15	2.76	6	40.00	9	60.00	
Living arrangements							0.168
Alone	80	14.73	28	35.00	52	65.00	
With partner	238	43.83	111	46.64	127	53.36	
2+ generations	212	39.04	104	49.06	108	50.94	
Other	13	2.39	7	53.85	6	46.15	
Living area							0.239
Rural area	104	19.15	42	40.38	62	59.62	
Urban area	439	80.85	208	47.38	231	52.62	
Employment status							0.325
Not employed	441	81.22	208	47.17	233	52.83	
Employed	102	18.78	42	41.18	60	58.82	
Length of membership ^※^	3.55 ± 3.12		3.52 ± 3.24		3.57 ± 3.02		0.871
Participation frequency							**0.037**
3–4 days weekly	49	9.02	16	32.65	33	67.35	
1–2 days weekly	351	64.64	158	45.01	193	54.99	
1–2 days monthly	143	26.34	76	53.15	67	46.85	
Number of club memberships							**<0.001**
1	323	59.48	168	52.01	155	47.99	
2+	220	40.52	82	37.27	138	62.73	
Travel time							0.685
<10 min	222	41.04	97	43.69	125	56.31	
10–20	207	38.26	99	47.83	108	52.17	
>20 min	112	20.70	52	46.43	60	53.57	
Time of visiting the CSC *							
9:00–12:00	372	68.51	176	47.31	196	52.69	0.433
12:00–15:00	110	20.26	41	37.27	69	62.73	**0.039**
After 18:00	77	14.18	33	42.86	44	57.14	0.630
Other	27	4.97	12	44.44	15	55.56	1.000
Satisfaction with sports service							**<0.001**
Satisfied	440	81.03	185	42.05	255	57.95	
Not satisfied	103	18.97	65	63.11	38	36.89	
Service quality for sports ^※^							
Benefits	12.60 ± 1.83		11.77 ± 1.77		13.31 ± 1.56		**<0.001**
Access	8.37 ± 1.45		7.95 ± 1.32		8.73 ± 1.47		**<0.001**
Interaction	13.13 ± 1.77		12.45 ± 1.78		13.71 ± 1.54		**<0.001**
Management	12.78 ± 1.84		12.15 ± 1.77		13.32 ± 1.72		**<0.001**

Significant *p*-values are shown in bold. * CSC = community sports club. ^※^ mean ± SD.

**Table 2 ijerph-17-08313-t002:** Multiple logistic regression model for factors associated with healthy active aging.

	Model I a			Model II b		
	Odd Ratio OR	95%		Odds Ratio (OR)	95%	
		Upper	Lower		Upper	Lower
Living arrangements ^※^						
Alone	2.385 *	1.282	4.439	2.681 *	1.315	5.467
With partner	1.073 *	0.701	1.644	1.417 *	0.828	2.425
2+ generations	1.000			1.000		
Educational level ^※^						
Junior high	1.000					
Senior high	1.310	0.686	2.503			
College	2.331 *	1.147	4.739			
Number of club memberships						
1	1.000			1.000		
2+	1.680 *	1.123	2.514	2.161 *	1.332	3.506
Service quality for sports						
Benefits	1.635 *	1.427	1.873	1.456 *	1.237	1.713
Management	1.234 *	1.091	1.394	1.269 *	1.097	1.469

* *p*-value < 0.05. ^※^ “other” group does not appear in this table. a: Participants over 60 years old, adjusted for living arrangements, educational level, number of club memberships, benefits, and management domains. b: Participants over 65 years old, adjusted for living arrangements, number of club memberships, benefits, and management domains.

**Table 3 ijerph-17-08313-t003:** Multiple logistic regression model for factors associated with healthy active aging, stratified by age group.

	Model I 60–64, *n* = 178			Model II 65–74, *n* = 269			Model III 75+, *n* = 96		
	Odds Ratio (OR)	95%		Odds Ratio (OR)	95%		Odds Ratio (OR)	95%	
		Upper	Lower		Upper	Lower		Upper	Lower
Living arrangements ^※^									
Alone				4.549 *	1.758	11.77			
With partner				1.297 *	0.673	2.501			
2+ generations				1.000 *					
Participation frequency ^※^									
3–4 days weekly							1.400 *	0.282	6.947
1–2 days weekly							1.000 *		
1–2 days monthly							4.061 *	1.018	16.203
Number of club memberships									
1				1.000 *					
2+				2.189 *	1.213	3.950			
Service quality for sports									
Benefits	1.728 *	1.342	2.227	1.659 *	1.344	2.047			
General	1.942 *	1.336	2.824						
Management				1.328 *	1.098	1.605	1.273 *	1.011	1.603

* *p*-value < 0.05. ^※^ “other” group does not appear in this table.

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
