# Peer review of "Service Quality for Sports and Active Aging in Japanese Community Sports Clubs"

_ijerph, 2020, doi:10.3390/ijerph17228313_

Round 1

Reviewer 1 Report

General comments:

The authors of this paper primarily explored the strength of association between service quality for sports and confounding healthy active aging attitude factors in a cross-sectional of 545 senior Japanese community sports club (CSC) members over 60 years old, recruited from across eight CSCs in Japan between 2012–2013. Such study is worthwhile in improving the quality of the sports environment for actively aging older people, often neglected in sports research and health domain.

The paper is well-written and methodologically sound, with good presentation of results and scientific discussion of the findings.

Specific comments:

I have few minor comments and corrections as follows:

Line: 31: ‘If we view the’… Rephrase this.

Line 34-35: ‘The idea of active aging 34 is to live in the present rather than fearing mortality and disability; thus, embracing a positive attitude is important in later life [8-9]’. The word of ‘fearing mortality and disability’ is not clear. Again, what do the authors mean by ‘in later life’. How later is later life, if the study is concerning the elderly?

Line 43-44: “…lacking human resources, and failures in administration cause unstable management in community sports organizations, causing elderly members to participate in sport. I suggest you rephrase this sentence─…lack of human resources, and administrative failures result to unstable ….causing inconsistence in sports participation by the elderly.

Line 87: ‘oft-‘ Correct the modern English usage to ‘often’.

Line 95: The ‘they’ in the ending of the sentence should be substituted with another better word. I suggest …’afterwards it decline.’

Line 98: ‘…status remains undeveloped…’ I suggest changing the word undeveloped’ to ‘unexplored’.

Line 153-154: ‘However, the association between service quality in sports and a positive aging attitude stratified by age group remains under discussion.’ This statement ending with ‘under discussion’ is not clear to the reader. Rephrase it.

Author Response

Thank you very much for your suggestions, we have modified in this revision according to reviewer's comments. Please see the attachment.

Point 1:

Line: 31: ‘If we view the’… Rephrase this.

Response 1: Thank you for your suggestion. We have revised the statements in the revision (Lines 31–33, page 1).

“If the concept of successful aging has been considered as a multidimensional but fundamental indicator of health, then active aging could be considered an advanced indicator for understanding health.”

Point 2:

Line 34-35: ‘The idea of active aging 34 is to live in the present rather than fearing mortality

and disability; thus, embracing a positive attitude is important in later life [8-9]’. The word of

‘fearing mortality and disability’ is not clear. Again, what do the authors mean by ‘in later life’. How later is later life, if the study is concerning the elderly?

Response 2: Thank you for your suggestion. We have revised the statements in the revision (Lines 34–35, page 1).

“The idea of active aging is to live in the present rather than concerning for onset of mortality and disability; thus, embracing a positive attitude is important in aging”

Point 3:

Line 43-44: “…lacking human resources, and failures in administration cause unstable management in community sports organizations, causing elderly members to participate in sport. I suggest you rephrase this sentence─…lack of human resources, and administrative failures result to unstable ….causing inconsistence in sports participation by the elderly.

Response 3: Thank you for your suggestion. We have revised the statements in the revision (Lines 43–46, pages 1–2).

“However, unstable financial support, lack human resources, and administrative failures result to unstable management in community sports organizations, causing inconsistence in sports participation by the elderly [15] or experience poor health outcomes [16].”

Point 4:

Line 87: ‘oft-‘ Correct the modern English usage to ‘often’.

Response 4: Thank you for your suggestion. We have revised the word in the revision (Line 88, page 2).

“Advanced age is another often mentioned demographic characteristic that influences customer satisfaction among patients of hospitals and clinics [37-38].”

Point 5:

Line 95: The ‘they’ in the ending of the sentence should be substituted with another better

word. I suggest …’afterwards it decline.’

Response 5: Thank you for your suggestion. We have revised the statements in the revision (Line 96, page 3).

“However, the high satisfaction scores only increase between ages 65 to 80, after which afterwards it decline.”

Point 6:

Line 98: ‘…status remains undeveloped…’ I suggest changing the word undeveloped’ to

‘unexplored’.

Response 6: Thank you for your suggestion. We have revised the word in the revision (Line 99, page 3).

“The relationship between service quality, satisfaction, and health status is widely discussed in healthcare system fields, but the relationship between service quality for sports and positive aging status remains unexplored in both the sports and health industry field.”

Point 7:

Line 153-154: ‘However, the association between service quality in sports and a positive aging attitude stratified by age group remains under discussion.’ This statement ending with ‘under discussion’ is not clear to the reader. Rephrase it.

Response 7: Thank you for your suggestion. We have revised the statements in the revision (Lines 155–156, page 4).

“However, the association between service quality in sports and a positive aging attitude stratified by age group remains unclear.”

Reviewer 2 Report

Congratulations on the argumentation and justification. Mainly in the introduction and discussion sections.

A great knowledge of the subject is denoted. The large number of references throughout the manuscript is highlighted. Congratulations for the good job in the validation of the different self-administered questionnaires.

However, they highlight some aspects that would need more detail. As for example at a methodological level. Below I detail the most representative considerations.

The authors report the date of the intervention in the year 2012-2013. Any reason that justifies the time that has passed?

All the variables analysed are from self-administered questionnaires. Do you have objective data on quality improvement? For example, a test of physical condition, cardiovascular or strength linked to improving quality of life.

Sorry,I could not find enough information regarding the collection of the questionnaires. For example, how they implemented the tests (by phone, in sports centres, ...).

In my opinion Figure 1, line 295, should appear before discussion

Author Response

Thank you very much for your suggestions, we have modified in this revision according to reviewer's comments. Please see the attachment for more details.

Point 1:

The authors report the date of the intervention in the year 2012-2013. Any reason that justifies the time that has passed?

Response 1: Thank you for your question. Since 2002, the Japanese government has focused on developing lifelong sports societies for people of all ages, genders, and disabilities; in addition, the government has implemented policies by establishing community sports clubs (CSCs) in every community. The government’s main goal is to increase the percentage of sports engagement by providing convenient and affordable places for sport. (Ministry of Education, Culture, Sport Science and Technology, MEXT 2020). The number of CSCs has achieved great growth year-on-year after a decade of this policy being implemented. The number and established percentage of CSCs in 2002, 2013, and 2019 were 541 (13.1%), 3,493 (79.0%), and 3,604 (80.5%), respectively (Figure 1). The development of policy implementation showed a growth trend in 2002–2012 but a slowdown in 2014–2019. The impact of policy on sports participation requires medium-term analysis to observe the change in individuals and communities in a five-to-ten-year period (Harris and Houlian, 2016). Therefore, the data collection in 2012–2013 not only represents the situation of CSCs in both study sites in Japan; more importantly, the 2012–2013 study year provides feedback for the decade-later impact of sport participation in communities, indicating a significant association between high-quality sports services and active aging for senior CSC members.  

Lin and Sakuno (2015) demonstrated how community sports environments provide sports services for senior customers and how older people engage in sports activities with an emphasis on urban–rural differences. The results suggest that the best service provider in communities, CSCs, can deliver quality sports services for a growing segment of the population. To our best knowledge, this study found an innovative view for examining the association between sports management and gerontology according to extensively many literature reviews. Furthermore, Shimizu and Yanagisawa (2015) investigated that 47.9% of CSCs in total developed in either a maintaining or shrinking trend in 2014, as evaluated using the predictor of memberships, financial resources, and number of employees and sports instructors. The results showed that the stable organization evolutionary growth of CSCs achieved a mature stage, which refers to a plateau phase of the growth curve. Our findings demonstrated the impact of community sports management on senior’s health status. Therefore, we believe that this study is worthwhile to publish in the International Journal of Environmental Research and Public Health according to its innovative idea about confirming the impact of sports participation in communities and improving the quality of sports environments for actively aging older people, who are often neglected in sports research and the health domain.

Point 2:

All the variables analysed are from self-administered questionnaires. Do you have objective data on quality improvement? For example, a test of physical condition, cardiovascular or strength linked to improving quality of life.

Response 2: Thank you for your question. Trends of healthy and active aging studies demonstrate a dynamic change from physiological constructs for active aging to comprehensively dimensions of mental and self-perceived health (Jaspers et al., 2017). Demographic differences among older individuals encourage policymakers to properly attend to the needs of the aging population. Certain characteristics, including young-old age, male gender, no disability, and high education level, are most often correlated with active aging [45-47]; however, the relationships exist under an objective rather than subjective interpretation of successful or healthy active aging [48]. (Lines 126–130, page 3). Suffering from diseases or functional impairment is not the only factor representing healthy active aging. For example, short form 36 items, as a measurement of health-related quality of life, has widely applied to evaluate health status in general or specific population with more than 130 diseases and conditions in the worldwide (Ware, 2000). These studies have shown a stable result on measuring the outcomes of different treatments, screening tool, and clinical and social interventions (Mulasso et al., 2014; Burholt et al., 2011). Subjective age or health and self-rated health have been utilized as indicators of healthy active aging [49-50] (Line 131–132, page 3). The self-administered questionnaires were utilized and collected in this study in order to clarify the positive effect of high quality of sports service on health active agers through the CSCs in community. The other results or methods were shown in a serial study, demonstrating the crucial role of managing sport resources by CSCs in both urban–rural areas (Lin & Sakuno, 2015). However, the limitation of this study or serial study should be highlighted for accuracy. Therefore, we have modified and explained further in the limitation of this revision according to reviewer’s suggestion (Lines 513–525, page 15).

“This study has several limitations. First, it is only a cross-sectional study with a purposive convenience sample of 545 participants in Tokyo and its neighboring area, which may limit the interpretability of the causality between service quality for sports and healthy active aging. Our findings confirmed the stratified effect of age segmentation on the association between service quality for sports and healthy active aging, but the representation is still warranted. Second, the study focused on Tokyo and Yamanashi prefecture in eastern Japan, which raises the issue of the findings’ generalizability. Thus, future studies should examine the active aging model and service quality in other Japanese communities and in other Asian countries with similar sociocultural backgrounds. Finally, this study utilized a self-designed questionnaire to represent healthy active aging rather than using an objective assessment tool such as senior fitness tests or history of chronic disease, which may have improved the accuracy. Furthermore, it is necessary to establish a model of active aging among Asians using both qualitative and quantitative approaches and develop a supportive sports environment for the elderly.”

Point 3:

Sorry, I could not find enough information regarding the collection of the questionnaires. For example, how they implemented the tests (by phone, in sports centres, ...).

Response 3: Thank you for your question. We have elaborated and explained further in the method of this revision (Lines 163–168, page 4).

“With the permission of the Tokyo governmental unit, this study delivered a questionnaire to five CSC leaders. Then, after completing the questionnaire, the CSCs returned it to the Tokyo Bureau of Sports, who in turn forwarded it to the author. For the three CSCs who agreed to participatory observation, the author acted as a sports volunteer and directly observed the operations of the CSCs for a month. Then, the questionnaires were delivered at the end of each sports program the following week in CSC. The final total was 545 participants over 60 years old from eight CSCs. The response rate was 70.32% (545/775 members).”

Point 4:

In my opinion Figure 1, line 295, should appear before discussion

Response 4: Thank you for your suggestion. We have modified the place of Figure 1 in the revision according to reviewer’s suggestion (Lines 301–331, page 10).

Reviewer 3 Report

This manuscript is well written, and the topic is interesting. It presents a good introduction and an adequate treatment of the analyzed data. It also presents a very interesting discussion.

However, there are very deficient aspects on which the whole study is based. The study has a convenience small sample.

The main problem I observe is the classification of active aging; it is based on a questionnaire of its own. I encourage you to publish the validation of this scale.

Furthermore, this study was developed seven years ago, and the elements and circumstances may have changed. For this reason, in my opinion, this manuscript should not be accepted for publication in this journal. I am very sorry for this decision. I encourage you to continue working on this topic and to improve the methodology of the study. It is a very well written and analyzed work.

Author Response

Thank you very much for your suggestions, we have modified in this revision according to reviewer's comments. Please see the attachment for more details.

Point 1:

This manuscript is well written, and the topic is interesting. It presents a good introduction and an adequate treatment of the analyzed data. It also presents a very interesting discussion. However, there are very deficient aspects on which the whole study is based. The study has a convenience small sample.

Response 1:

Thank you for your question. We provided further evidences and explanations to support our data and reflect the representation in the following statements.

Firstly, this cross-sectional study used a convenience sample of 545 senior Japanese CSC members over 60 years old, recruited from across eight CSCs in Nerima, a ward in Tokyo, and Otsuki, a city in Yamanashi Prefecture, Japan. The two study sites were selected for several reasons. First, regarding the purposes of this study, the study sites were selected according to the following regional characteristics: the population, the percentage of older people, established percentage of CSCs, and the number of CSCs in community (Table 1). Second, administrative support was sought from local governing bodies in order to conduct the study smoothly. Consultation meetings with local government representatives from sports organizations were conducted face-to-face. With 709,262 citizens, Nerima ward was chosen because the Tokyo government had selected Nerima as the model for developing CSCs; thus, its practices exemplified the desired practice for other sports clubs in Japan. Nerima ward contains 7 CSCs and 675 senior members over 60 years old. Otsuki city, with 27,505 citizens, was selected as the remote site per the recommendation of the Tokyo government due to the population and number of senior members. Otsuki city contains only 1 CSC and 106 senior members aged above 60 years old.

Secondly, power analysis was conducted using G*Power 3.1.9.2, with the significance level α = 0.05, statistical power 0.95, and R2 power of 0.16. The analysis estimated that the required sample size was 411 (Faul et al., 2007). On the other hand, according to the formula for calculating sample size with a 95% confidence level, 4.2 confidence interval, and 736,767 population in Nerima ward (709,262) and Otsuki city (27,505), a 544 sample size was needed (Cohen, 1988). This study collected 545 participants, which fit the minimum sample size for both estimations.

Finally, we investigated the latest three-year published papers in the International Journal of Environmental Research and Public Health, which included the keyword of “the elderly” shown in the title through the database of “PubMed” during 2018–2020. In total, 59 journal articles were searched. Among them, eight utilized a convenient sampling with reasonable samples from 154–545 participants (Sun et al., 2020; Lara et al., 2020; Rodríguez-Guerrero et al., 2020; Chen et al., 2020; Tornero-Quiñones et al., 2020; Król-ZieliÅ„ska et al., 2019; Choi et al., 2019; Fragakis et al., 2018), four articles utilized data analysis during 2013–2015 (Mihailovic et al., 2020; Pan et al., 2019; Zhang et al., 2018; Wang et al., 2019), and three published articles had an unknown study period (Hong et al., 2020; Tornero-Quiñones et al., 2020; Chen et al., 2020). These articles have been published by the journal due to their innovation ideas, diversity in culture, and clearly valuable content for worldwide readers and scholars.

We really appreciate for your question; therefore, we have modified and explained further in the limitation of this revision according to reviewer’s suggestion (Lines 513–525, page 15).

“This study has several limitations. First, it is only a cross-sectional study with a purposive convenience sample of 545 participants in Tokyo and its neighboring area, which may limit the interpretability of the causality between service quality for sports and healthy active aging. Our findings confirmed the stratified effect of age segmentation on the association between service quality for sports and healthy active aging, but the representation is still warranted. Second, the study focused on Tokyo and Yamanashi prefecture in eastern Japan, which raises the issue of the findings’ generalizability. Thus, future studies should examine the active aging model and service quality in other Japanese communities and in other Asian countries with similar sociocultural backgrounds. Finally, this study utilized a self-designed questionnaire to represent healthy active aging rather than using an objective assessment tool such as senior fitness tests or history of chronic disease, which may have improved the accuracy. Furthermore, it is necessary to establish a model of active aging among Asians using both qualitative and quantitative approaches and develop a supportive sports environment for the elderly.”

Point 2:

The main problem I observe is the classification of active aging; it is based on a questionnaire of its own. I encourage you to publish the validation of this scale.

Response 2: Thank you for your suggestion. We have provided the validation of the scale in the previous manuscript (Lines 188–189, page 4) and more detailed in appendix A (Lines 762–780, pages 21-22) this revision according to reviewer’s suggestion. 

The Health and aging scale was developed for this study to examine the healthy active aging status of older people. Through confirmatory factor analysis, four factors of health status were identified. For the Health and aging scale, confirmatory factor analysis of the revised scale (11 items, 4 factors) produced good fit indices (RMR = .04; GFI = .93; AGFI = .88; DELTA2 = .91; CFI = .91; RMSEA = .09; χ2 =234.47 ; df=38) (Table 2). Items 5, 6, and 14 were deleted due to their low R2 values in the original version (Lin, 2014). The revised version of questionnaire was used to analyse and the internal consistency estimates revealed an overall alpha of .81. For the subscales, alpha values were .85 for psychological health, .68 for social health, .62 for positive attitude, and .72 for morale (Table 3).

Point 3:

Furthermore, this study was developed seven years ago, and the elements and circumstances may have changed. For this reason, in my opinion, this manuscript should not be accepted for publication in this journal. I am very sorry for this decision. I encourage you to continue working on this topic and to improve the methodology of the study. It is a very well written and analyzed work.

Response 3: Thank you for your question. We provided further evidences and explanations to support our data and reflect the representation in the following statements.

Since 2002, the Japanese government has focused on developing lifelong sports societies for people of all ages, genders, and disabilities; in addition, the government has implemented policies by establishing community sports clubs (CSCs) in every community. The government’s main goal is to increase the percentage of sports engagement by providing convenient and affordable places for sport. (Ministry of Education, Culture, Sport Science and Technology, MEXT 2020). The number of CSCs has achieved great growth year-on-year after a decade of this policy being implemented. The number and established percentage of CSCs in 2002, 2013, and 2019 were 541 (13.1%), 3,493 (79.0%), and 3,604 (80.5%), respectively (Figure 1). The development of policy implementation showed a growth trend in 2002–2012 but a slowdown in 2014–2019. The impact of policy on sports participation requires medium-term analysis to observe the change in individuals and communities in a five-to-ten-year period (Harris and Houlian, 2016). Therefore, the data collection in 2012–2013 not only represents the situation of CSCs in both study sites in Japan; more importantly, the 2012–2013 study year provides feedback for the decade-later impact of sport participation in communities, indicating a significant association between high-quality sports services and active aging for senior CSC members.  

Lin and Sakuno (2015) demonstrated how community sports environments provide sports services for senior customers and how older people engage in sports activities with an emphasis on urban–rural differences. The results suggest that the best service provider in communities, CSCs, can deliver quality sports services for a growing segment of the population. To our best knowledge, this study found an innovative view for examining the association between sports management and gerontology according to extensively many literature reviews. Furthermore, Shimizu and Yanagisawa (2015) investigated that 47.9% of CSCs in total developed in either a maintaining or shrinking trend in 2014, as evaluated using the predictor of memberships, financial resources, and number of employees and sports instructors. The results showed that the stable organization evolutionary growth of CSCs achieved a mature stage, which refers to a plateau phase of the growth curve. Our findings demonstrated the impact of community sports management on senior’s health status. Therefore, we believe that this study is worthwhile to publish in the International Journal of Environmental Research and Public Health according to its innovative idea about confirming the impact of sports participation in communities and improving the quality of sports environments for actively aging older people, who are often neglected in sports research and the health domain.

Secondly, the criteria of study site were selected according the population, the percentage of the elderly population, established percentage of CSCs, and the number of CSCs in community (see reply to point 1). We investigated the current situation of study sites with the dimensions of nation, region/area, and community, and compared the changes of number of CSC, established percentage of CSCs, population of older people, and percentage of elderly population in 2013 and 2019 (Table 4). In Japan, the number and established percentage of CSCs were 3,493 and 79.0 % in 2013 (3,604 and 80.5 % in 2019, respectively). In Tokyo, percentage of older people, the number and established percentage of CSCs were 20.82 %, 48 CSCs, and 77.42% (22.07 %, 58, and 93.54%). In Yamanashi prefecture, percentage of older people, the number and established percentage of CSCs were 25.69 %, 23 CSCs, and 85.19% (29.38 %, 22, and 81.48%). In 2013, Nerima ward contained 7 CSCs and 20.28 % of older people (7 and 21.78 % in 2019); and Otsuki city contained 1 CSC and 31.26 % of older people (1 and 37.60 % in 2019). According to the compared results, population of older people has gradually increased; however, the development of number of CSCs or established percentage of CSCs in nation, region/area, and community remains stable during 2013-2019.

Finally, we investigated the latest three-year published papers in the International Journal of Environmental Research and Public Health, which included the keyword of “the elderly” shown in the title through the database of “PubMed” during 2018–2020. In total, 59 journal articles were searched. Among them, eight utilized a convenient sampling with reasonable samples from 154–545 participants, four articles utilized data analysis during 2013–2015, and three published articles had an unknown study period (see reply to point 1). These articles have been published by the journal due to their innovation ideas, diversity in culture, and clearly valuable content for worldwide readers and scholars.

Therefore, we believe this study is worthwhile to publish in the International Journal of Environmental Research and Public Health according to its innovating idea in confirming the impact of sport participation in community and improving the quality of the sports environment for actively aging older people, often neglected in sports research and health domain.

Round 2

Reviewer 2 Report

Thank very much to the author for the quality answer to the questions. After reading, just a small consideration. Please, could you include the evolution of CSCs to help to the readers?

Author Response

Point 1:

Thank very much to the author for the quality answer to the questions. After reading, just a small consideration. Please, could you include the evolution of CSCs to help to the readers?

Response 1:

Thank you for your suggestion. We have modified and separated the paragraph into the introduction (Lines 104–107, 126–142, page 3), discussion (Lines 340–357, page 11), and references (Lines 695–709, page 18) in this revision according to the reviewer’s suggestions.

1. Introduction (Lines 104–107, 126–142, page 3)

1.2. The importance of community sports clubs for older Japanese people

……The main goals of the CSC project include providing convenient and affordable sports complexes for all residents in a community and creating a sports environment accessible to residents of all ages, genders, and ability levels, thus facilitating social activities. The number of CSCs has achieved great growth year-on-year after a decade of this policy being implemented. The number and established percentage of CSCs in 2002, 2013, and 2019 were 541 (13.1%), 3,493 (79.0%), and 3,604 (80.5%), respectively (Figure 1). [41]. In terms of age, members under 12 years old constitute the largest membership group (19.6%); older participants include those aged 70+ years (10.9 %) and adults aged 60–69 (9.5%). The number of 60+ participants will continue to rise as the population ages…….

4. Discussion (Lines 340–357, page 11)

CSC memberships have reached 3,604 CSCs during 2012–2019. The development of policy implementation showed a growth trend in 2002–2012 but a slowdown in 2014–2019. The impact of policy on sports participation requires medium-term analysis to observe the change in individuals and communities in a five-to-ten-year period [64]. Therefore, the data collection in 2012–2013 not only represents the situation of CSCs in both study sites in Japan; more importantly, the 2012–2013 study year provides feedback for the decade-later impact of sport participation in communities, indicating a significant association between high-quality sports services and active aging for senior CSC members. Lin and Sakuno (2015) demonstrated how community sports environments provide sports services for senior customers and how older people engage in sports activities with an emphasis on urban–rural differences. The results suggest that the best service provider in communities, CSCs, can deliver quality sports services for a growing segment of the population [65]. To our best knowledge, this study found an innovative view for examining the association between sports management and gerontology according to extensively many literature reviews. Furthermore, Shimizu and Yanagisawa (2015) investigated that 47.9% of CSCs in total developed in either a maintaining or shrinking trend in 2014, as evaluated using the predictor of memberships, financial resources, and number of employees and sports instructors [66]. The results showed that the stable organization evolutionary growth of CSCs achieved a mature stage, which refers to a plateau phase of the growth curve.

In this community cohort study, the proportion of subjective HA was 53.96%, which resembled that reported in a previous study conducted in European countries (56.2%) and worldwide (0.4%–95%) [45, 67]. In the present study, factors including living alone, being a college graduate, having more than one club membership, and highly evaluating service quality for sports were associated with HA…….

Reviewer 3 Report

I would like you to include in the manuscript the explanation you sent me regarding the evolution of CSCs from 2002 to 2019. This would make it clear to future readers why the 2013 data may be similar to the current data.

Other comments:

line 242 (page 6): revise data about college graduates (different from the table)

Author Response

Point 1:

I would like you to include in the manuscript the explanation you sent me regarding the evolution of CSCs from 2002 to 2019. This would make it clear to future readers why the 2013 data may be similar to the current data.

Response 1:

Thank you for your suggestion. We have modified and separated the paragraph into the introduction (Lines 104–107, 126–142, page 3), discussion (Lines 340–357, page 11), and references (Lines 695–709, page 18) in this revision according to the reviewer’s suggestions.

1. Introduction (Lines 104–107, 126–142, page 3)

1.2. The importance of community sports clubs for older Japanese people

……The main goals of the CSC project include providing convenient and affordable sports complexes for all residents in a community and creating a sports environment accessible to residents of all ages, genders, and ability levels, thus facilitating social activities. The number of CSCs has achieved great growth year-on-year after a decade of this policy being implemented. The number and established percentage of CSCs in 2002, 2013, and 2019 were 541 (13.1%), 3,493 (79.0%), and 3,604 (80.5%), respectively (Figure 1). [41]. In terms of age, members under 12 years old constitute the largest membership group (19.6%); older participants include those aged 70+ years (10.9 %) and adults aged 60–69 (9.5%). The number of 60+ participants will continue to rise as the population ages…….

4. Discussion (Lines 340–357, page 11)

CSC memberships have reached 3,604 CSCs during 2012–2019. The development of policy implementation showed a growth trend in 2002–2012 but a slowdown in 2014–2019. The impact of policy on sports participation requires medium-term analysis to observe the change in individuals and communities in a five-to-ten-year period [64]. Therefore, the data collection in 2012–2013 not only represents the situation of CSCs in both study sites in Japan; more importantly, the 2012–2013 study year provides feedback for the decade-later impact of sport participation in communities, indicating a significant association between high-quality sports services and active aging for senior CSC members. Lin and Sakuno (2015) demonstrated how community sports environments provide sports services for senior customers and how older people engage in sports activities with an emphasis on urban–rural differences. The results suggest that the best service provider in communities, CSCs, can deliver quality sports services for a growing segment of the population [65]. To our best knowledge, this study found an innovative view for examining the association between sports management and gerontology according to extensively many literature reviews. Furthermore, Shimizu and Yanagisawa (2015) investigated that 47.9% of CSCs in total developed in either a maintaining or shrinking trend in 2014, as evaluated using the predictor of memberships, financial resources, and number of employees and sports instructors [66]. The results showed that the stable organization evolutionary growth of CSCs achieved a mature stage, which refers to a plateau phase of the growth curve.

In this community cohort study, the proportion of subjective HA was 53.96%, which resembled that reported in a previous study conducted in European countries (56.2%) and worldwide (0.4%–95%) [45, 67]. In the present study, factors including living alone, being a college graduate, having more than one club membership, and highly evaluating service quality for sports were associated with HA…….

Point 2:

Line 242 (page 6): revise data about college graduates (different from the table)

Response 2:

Thank you for the reminder. It was a typing error. We have carefully checked and modified the number throughout the manuscript in this revision (Line 262, page 6).
